# Prevalence and factors associated with contraceptive use among sexually active adolescent girls in 25 sub-Saharan African countries

**Turnwait Otu Michael** [1,2]*, **Tolulope Funmilola Ojo**[3], **Olasupo Augustine Ijabadeniyi**[4], **Michael Ayodele Ibikunle**[4], **James Olukayode Oni**[5], **Adebanke Adeorite Agboola**[6]

1 Department of Sociology, University of Ibadan, Ibadan, Nigeria, 2 Department of Sociology, University of Johannesburg, Johannesburg, South Africa, 3 Department of Public Health, Afe Babalola University, Ado-Ekiti, Nigeria, 4 Department of Sociology, Afe Babalola University, Ado-Ekiti, Nigeria, 5 Department of Pharmacology and Therapeutics, Afe Babalola University, Ado-Ekiti, Nigeria, 6 Department of Integrated Medical Sciences, Afe Babalola University, Ado-Ekiti, Nigeria

* turnwaitmichael@gmail.com

**Data Availability Statement:** The data used for this study are publicly available from The DHS Program. Interested researchers can register on

## Abstract

### Introduction

Various countries in sub-Saharan Africa have taken divergent steps toward achieving the Sustainable Development Goal's target of universal access to sexual and reproductive health-care services by 2030, particularly among sexually active adolescent girls who are at risk of unplanned pregnancies and sexually transmitted infections. However, because contraceptive use among sexually active adolescents remains unexplored in sub-Saharan Africa, the researchers intended to examine the prevalence and factors associated with contraceptive use among adolescent girls who had been sexually active in the previous four weeks.

### Materials and methods

Cross-sectional data from the most recent demographic and health surveys of 25 sub-Saharan African countries on 16,442 sexually active adolescent girls were analyzed. In the analyses, descriptive statistics and multivariate binary logistic regression were used. Analyses were statistically significant at p<0.05.

### Results

The overall prevalence of contraceptive use was 25.4%. Chad had the lowest prevalence (4%), while Namibia had the highest (60.5%). Over 90% of the countries studied had less than 50% contraceptive use among sexually active adolescent girls. Adolescent girls with higher education were eight times more likely than those with no formal education to use contraception (aOR = 7.97, 95% CI = 6.26-9.45). When compared to single adolescent girls, married adolescent girls were 66% less likely to use contraceptives (aOR = 0.34, 95% CI =

the DHS Program website (https://dhsprogram.
com/) and subsequently request access to the
datasets. The authors confirm that they do not
have any special access privileges.

**Funding:** The author(s) received no specific
funding for this work.

**Competing interests:** The authors have declared
that no competing interests exist.

0.31-0.36). Adolescent girls with two or more children were seven times more likely than
those without a child to use contraceptives (aOR = 6.91, 95% CI = 5.58-8.56).

## Conclusion

It is established that there exists a low prevalence of contraceptive use among adolescent
girls in sub-Saharan Africa. As countries in the sub-region strive for universal access to
reproductive health services, it is critical for the governments and civil societies in countries
with low contraceptive use to strengthen mass education on the use of contraception
among sexually active adolescents, with special emphasis on the less educated, married,
and adolescent girls from poor households.

## Introduction

Adolescents are young people aged 10 to 19 [1]. In 2022, over 13 million births occurred
among adolescents worldwide [2]. The Demographic and Health Survey collected information
on the sexual and reproductive health (SRH) of adolescent girls at the age of 15 [3]. Every year,
approximately 21 million girls between the ages of 15 and 19 become pregnant in developing
countries [1]. In 2021, about 12 million births occurred among adolescents in developing
countries. Adolescent girls in Africa account for 14% of all babies born in Africa, compared to
nine percent globally [4]. While adolescent birth rates have declined in other parts of the
world, sub-Saharan Africa (SSA) continues to have the highest (101 births per 1000 adolescent
girls, exceeding the world average of 45 births per 1,000 adolescent girls), owing to a high
unmet need for contraception among adolescents [4–6].

Contraception can be modern or traditional. Condoms, pills, implants, intra-uterine
devices, and injectables are among the modern contraceptives that are more reliable and effec-
tive in preventing unwanted pregnancies [7, 8]. Unreliable traditional contraceptives are with-
drawal, periodic abstinence, and herbal mixtures or concoctions [9, 10]. The World Health
Organization states that adolescents, like adults, are entitled to SRH rights, such as access to
counseling, contraception, and sex education [11]. Access to and use of modern contraceptives
among sexually active adolescents has been shown to promote sexual behavior while also
reducing maternal health risks, school dropout, and economic hardship in young girls [12].
Millions of adolescent girls, particularly in developing countries, lack access to contraception
[13]. This has put many adolescent girls at risk of unplanned pregnancy, unsafe abortion, and
maternal morbidity and mortality [11].

Though contraceptive use among adolescent girls varies by country, 14 million (43%) ado-
lescent girls in low- and middle-income countries (LMICs) have an unmet need for modern
contraception. Unintended pregnancies account for approximately 49% (10 million) of the 21
million pregnancies that occur among adolescents in LMICs each year; 50% of these pregnan-
cies result in unsafe abortion [14]. Over 50% of adolescents in SSA have unmet contraception
needs [15]. Babies born by adolescents who carried their pregnancies to term are 50% more
likely to die, be born prematurely, or have serious health complications than babies born by
older women [16–18].

Previous research had primarily focused on contraceptive use among adolescents in gen-
eral, particularly at the country level [19–23]. At the multi-country level, no attention has been
paid to sexually active adolescents' contraceptive use in SSA. A few multi-country level studies
that examined contraceptive use among adolescents in SSA focused only on married/cohabited

adolescents [24], young adult women aged 15 to 24 without disaggregation by adolescent age [25], and school-going adolescents (excluding out-of-school adolescents) [26]. The purpose of this study was to fill this void by assessing the prevalence and factors associated with contraceptive use among sexually active adolescent girls in 25 SSA countries, in order to provide empirical evidence for advocacy and comparative policy options.

## Materials and methods

### Data source

The source of this study's data was from Demographic and Health Surveys (DHS) conducted in 25 SSA countries between January 1, 2012, and December 31, 2021. Table 1 shows the countries that were chosen. Individual recode files (women) were used. The most recent surveys from the countries chosen were used. The surveys from the included countries were selected because they all contained the key variables of interest for this study. The DHS collected information on adolescent girls and women's sexual activity, contraception, fertility, individual and household socioeconomic, and demographic factors. DHS is a five-year national survey conducted in LMICs. DHS employs a multi-stage sampling technique in all countries, beginning

**Table 1. Sample size and year of survey.**

| Country | Year | Weighted Sample | Weighted % |
|---|---|---|---|
| 1) Angola | 2015-16 | 961 | 5.8 |
| 2) Benin | 2017-18 | 743 | 4.5 |
| 3) Burundi | 2016-17 | 230 | 1.4 |
| 4) Chad | 2014-15 | 1,164 | 7.1 |
| 5) Congo, DR* | 2013-14 | 1,075 | 6.5 |
| 6) Ethiopia | 2016 | 504 | 3.1 |
| 7) Gabon | 2012 | 612 | 3.7 |
| 8) Gambia | 2019-20 | 255 | 1.5 |
| 9) Ghana | 2014 | 228 | 1.4 |
| 10) Guinea | 2018 | 493 | 3.0 |
| 11) Kenya | 2014 | 311 | 1.9 |
| 12) Lesotho | 2014 | 182 | 1.1 |
| 13) Liberia | 2019-20 | 498 | 3.0 |
| 14) Madagascar | 2021 | 1,389 | 8.4 |
| 15) Malawi | 2015-16 | 1,180 | 7.2 |
| 16) Mali | 2018 | 789 | 4.8 |
| 17) Namibia | 2013 | 248 | 1.5 |
| 18) Niger | 2012 | 696 | 4.2 |
| 19) Nigeria | 2018 | 1,913 | 11.6 |
| 20) Rwanda | 2019-20 | 112 | 0.7 |
| 21) Sierra Leone | 2019 | 972 | 5.9 |
| 22) South Africa | 2016 | 241 | 1.5 |
| 23) Tanzania | 2015-16 | 700 | 4.3 |
| 24) Zambia | 2018 | 566 | 3.4 |
| 25) Zimbabwe | 2015 | 384 | 2.3 |
| **All Countries** | - | **16,442** | **100** |

Note:

*DR = Democratic Republic; DHS 2012-2021

with cluster sampling of enumeration areas (EAs), followed by systematic selection of house-holds, and simple random sampling of one eligible respondent from each household. This study sample (N = 16,442) consisted of sexually active adolescents aged 15 to19 who responded to having been sexually active in the four weeks preceding the survey and had complete cases of the concerned variables (S1 File). In writing this manuscript, the authors adhere to the 'Strengthening the Reporting of Observational Studies in Epidemiology' (STROBE) statement [27] as described in the S2 File.

## Study variables

**Outcome variable.** The use of contraceptives by adolescent girls who were sexually active was the study's outcome variable. Female and male sterilization, IUD, injectables, implants, pill, condom, female and male condoms, diaphragm, foam/jelly, lactational amenorrhea, rhythm, withdrawal, and no methods were the options available to respondents. To create a binary outcome variable for logistic regression, the contraceptive method categories were recoded as follows: no method = 0, any method = 1.

**Explanatory variables.** Education, marital status, working status, ideal number of chil-dren, total number of children, ever had a terminated pregnancy and knowledge of contracep-tive methods were the explanatory variables. Others were the number of sex partners, ability to request that a partner use a condom, ever been tested for HIV, wealth index, and type of resi-dence, as well as listening to family planning programme on radio, TV, newspaper/magazine in the last few months. Education was coded as no education = 0, primary education = 1, sec-ondary education = 2, and higher education = 3. The marital status was recoded as follows: never married = 0, ever married = 1. Working status was coded as 0 (not working) or 1 (work-ing). The ideal number of children was recoded as 0-2 = 1, 3-5 = 2, and 6 or more = 3. The total number of children born was recoded as none = 0, one = 1, two or more = 2. Never had a termination of pregnancy was coded as No = 0, Yes = 1. Family planning overheard on the radio was coded (No = 0, Yes = 1). Family planning heard on television was coded as No = 0 and Yes = 1. Family planning heard through the newspaper/magazine was coded as No = 0, Yes = 1. The number of sexual partners was recoded as one = 1, two or more = 2. The ability to request that a partner use a condom was coded as No = 0 and Yes = 1. Ever tested for HIV was coded as No = 0 and Yes = 1. Poorest = 1, Poorer = 2, Middle = 3, Richer = 4, and Richest = 5 were assigned to the wealth index. The location of residence was coded as Urban = 1 and Rural = 2. The DHS dataset's wealth index was computed in quintiles using Principal Compo-nent Analysis (PCA), capturing respondents' ownership of household assets such as bicycles, televisions, water sources, toilet facilities, and building materials [28]. The availability of the explanatory variables in the DHS datasets influenced their selection. A previous review of the literature on the relationship between factors associated with contraceptive use also supported variable selection [3, 18, 20].

## Statistical analyses

The statistical analysis was carried out in three steps. The first was the use of univariate analysis to describe sample and explanatory variables. The second was the use of bivariate analysis on explanatory variables against contraceptive use by methods (none, traditional, or modern) via cross tabulation and chi-square statistics. The chi-square test was accepted at p<0.05. Only variables that were significant in the chi-square test advanced to the final stage. Variance infla-tion factor (VIF) was used for collinearity diagnostics to check for multicollinearity among predictor variables. According to the diagnostic test results, none of the explanatory variables had collinearity: minimum VIF = 1.01 and maximum VIF = 1.37 (see S3 File). The reference

groups were chosen based on normative categories, the lowest likelihood of using any contraception methods as documented in the literature, and/or the largest category of observations. To avoid oversampling and under-sampling errors, the DHS datasets were weighted using the women's unit of analysis (v005) at "COMPUTE WGT = V005/1000000, WEIGHT by WGT." SPSS Statistics 25 software was used for the analysis, and all statistics were tested at a confidence level of 0.95.

### Ethical clearance

The authors obtained written authorization from the DHS Program to use DHS datasets in this study. The DHS received ethical approval from the ICF Institutional Review Board (ICF IRB FWA00000845) in the United States, as well as from the National Health Research Ethics Committees of the countries chosen for this study. In Nigeria, for example, the National Health Research Ethics Committee of Nigeria granted ethical approval for the survey (NHREC/01/01/2007). All participants provided written informed consent. In the case of minors under the age of 18, their parents or guardians provided informed consent. There was no need for further participation consent in this study because the research used DHS (secondary) datasets. More information about DHS ethical concerns can be found at https://goo.gl/ny8T6X. The DHS datasets are publicly accessible at https://dhsprogram.com/data/.

### Results

Fig 1 depicts the prevalence of contraceptive use among sexually active adolescent girls in SSA. Findings showed that the overall prevalence of contraceptive use in SSA was 25.4%, with rates ranging from four percent in Chad to 60.5% in Namibia. Chad (4%), Nigeria (5.4%), Niger (7.2%), and Gambia (8.2%) had lower than 10% contraceptive use. Lesotho (53.6%), South Africa (59.8%), and Namibia (60.5%) had more than 50% contraceptive use (See S1 File).

Table 2 presents the results regarding background characteristics and contraceptive use among sexually active adolescent girls in SSA. Sixty-four percent of adolescent girls had less than secondary education, 66.7% were married, 43.7% were working, 43.4% had at least one child, 6.9% had terminated a pregnancy in their lifetime, and 25.9%, 11.6%, and 5.0% heard about family planning on radio, television, and newspaper/magazine, respectively. In the previous 12 months, 10% of respondents had two or more sex partners, 33.9% had been tested for HIV, 22.3% came from the poorest households, and 69.3% lived in rural areas. The chi-square statistical tests revealed that all of the independent variables were significantly associated with adolescent girls' use of contraceptives (See S4 File).

The unadjusted and adjusted odds ratios of binary logistic regression of contraceptive use among adolescent girls in SSA are shown in Table 3. Education, marital status, ideal number of children, number of children ever born, heard family planning on radio, ability to request that a partner use a condom, ever been tested for HIV, wealth index, and residence were all significantly associated with contraceptive use among adolescent girls in the unadjusted logistic regression model. In the adjusted model, adolescent girls with at least secondary education were twice as likely as those with less than secondary education to use contraceptives. When compared to single adolescent girls, married adolescent girls were 66% less likely to use contraceptives. Adolescent girls with an ideal number of 6 or more children were 70% less likely to use contraceptives than those with an ideal number of 0-2 children. Adolescent girls with two or more children were seven times more likely than those without a child to use contraceptives. Adolescent girls who heard about family planning on the radio [aOR = 1.25, 95% Cl = 1.09-1.44] or read from the newspaper/magazine [aOR = 1.43, 95% Cl = 1.04-1.96] were more likely to use contraceptives than those who did not. Adolescent girls who could request that their

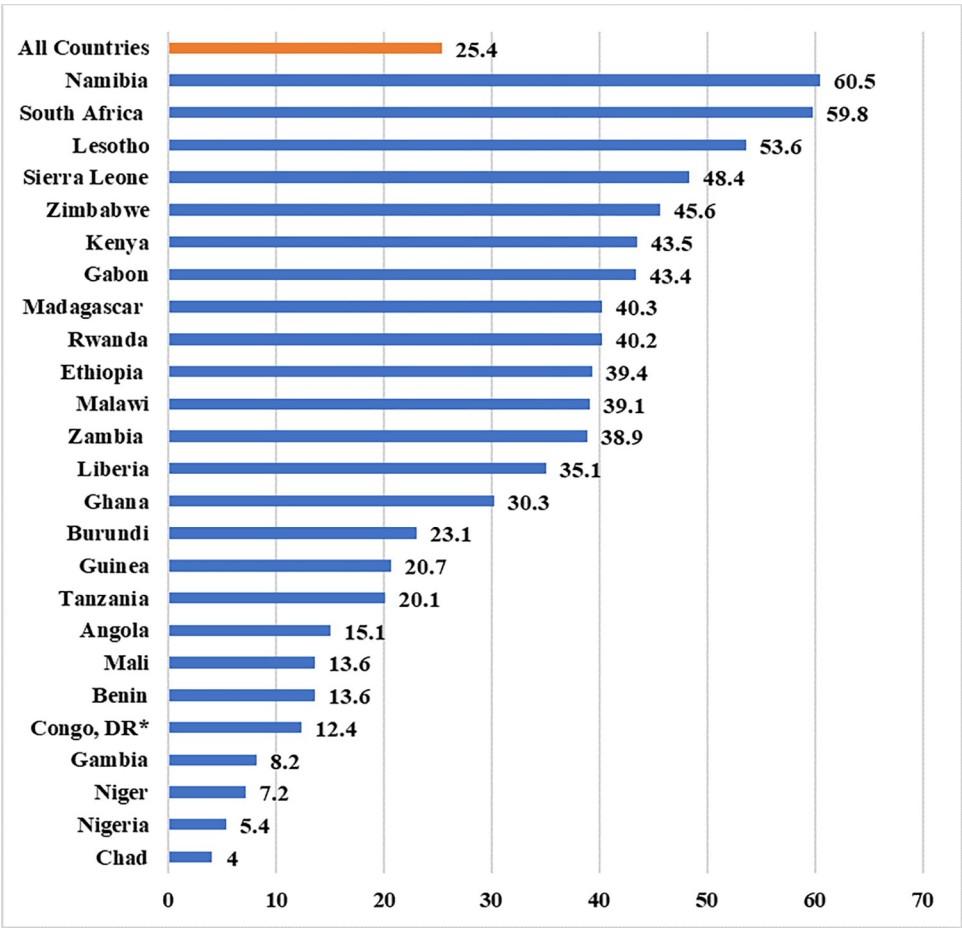

**Fig 1. Prevalence of modern contraceptive use among sexually active adolescent girls in sub-Saharan Africa.**

partners use a condom were twice as likely as those who could not. Girls who had ever been tested for HIV were twice as likely as those who had never been tested to use contraceptives. Others were more likely to use contraceptives than adolescent girls from the poorest households. Surprisingly, adolescent girls in rural areas were twice as likely as those in urban areas to use contraceptives (both modern and traditional methods combined) (see S5 File). However, as shown in Table 2, adolescent girls in urban areas (32.5%) were more likely to use modern contraception than those in rural areas (22.2%).

## Discussion

The investigation focused on the prevalence of contraceptive use among sexually active adolescent girls in 25 SSA countries. The overall prevalence of contraceptive use indicates 25.4%. While Namibia (60.5%) has the most prevalent use of contraceptives, Chad has the lowest prevalence (4%). Contraception use is lower than 10% in Chad (4%), Nigeria (5.4%), Niger (7.2%), and Gambia (8.2%). More than 50% of adolescent girls in Lesotho (53.6%), South Africa (59.8%), and Namibia (60.5%) make use of contraceptives. The lower prevalence of contraceptive use among sexually active adolescent girls in Chad is attributed to harmful cultural practices, humanitarian crises, a lack of reproductive health rights, a lack of family planning commodities, and a lack of funding for family planning in the government budget, despite the

**Table 2. Background characteristics and contraceptive use among sexually active adolescent girls in sub-Saharan Africa.**

| Variables | Weighted (N = 16,442) | | Contraceptive Method Type | | | |
|---|---|---|---|---|---|---|
| | N | % | None (%) | Traditional (%) | Modern (%) | p-values |
| **Education** | | | | | | <0.001 |
| No education | 4732 | 28.8 | 91.1 | 0.7 | 8.2 | |
| Primary | 5744 | 34.9 | 70.7 | 2.2 | 27.1 | |
| Secondary | 5854 | 35.6 | 57.4 | 5.6 | 37.1 | |
| Higher | 113 | 0.7 | 48.7 | 2.7 | 48.7 | |
| **Marital status** | | | | | | <0.001 |
| Never married | 5,469 | 33.3 | 56.3 | 5.3 | 38.4 | |
| Ever married | 10,973 | 66.7 | 79.3 | 1.8 | 18.9 | |
| **Working status** | | | | | | 0.947 |
| Not working | 9,235 | 56.3 | 71.7 | 2.9 | 25.4 | |
| Working | 7,178 | 43.7 | 71.5 | 3.0 | 25.5 | |
| **Ideal number of children** | | | | | | <0.001 |
| 0-2 | 1,998 | 12.7 | 56.9 | 3.0 | 40.1 | |
| 3-5 | 8,764 | 55.6 | 64.3 | 3.7 | 32.0 | |
| 6+ | 5,013 | 31.8 | 87.8 | 1.9 | 10.3 | |
| **Total children ever born** | | | | | | <0.001 |
| None | 9,307 | 56.6 | 75.5 | 3.3 | 21.2 | |
| 1 | 5,747 | 35.0 | 64.3 | 2.6 | 33.1 | |
| 2 or more | 1,388 | 8.4 | 76.3 | 2.2 | 21.5 | |
| **Ever had a terminated pregnancy** | | | | | | <0.001 |
| No | 15,295 | 93.1 | 71.3 | 2.9 | 25.9 | |
| Yes | 1,141 | 6.9 | 77.0 | 4.1 | 18.9 | |
| **Heard family planning on radio last few months** | | | | | | <0.001 |
| No | 12,177 | 74.1 | 74.0 | 2.8 | 23.2 | |
| Yes | 4,258 | 25.9 | 65.0 | 3.5 | 31.5 | |
| **Heard family planning on TV last few months** | | | | | | <0.001 |
| No | 14,534 | 88.4 | 73.2 | 2.7 | 24.1 | |
| Yes | 1,904 | 11.6 | 60.3 | 5.1 | 34.6 | |
| **Heard family planning in newspaper/magazine last few months** | | | | | | <0.001 |
| No | 15,628 | 95.0 | 72.6 | 2.9 | 24.6 | |
| Yes | 814 | 5.0 | 54.7 | 4.5 | 40.8 | |
| **Number of sex partners in last 12 months** | | | | | | <0.001 |
| 1 | 14,819 | 90.1 | 72.5 | 2.7 | 24.8 | |
| 2 or more | 1622 | 9.9 | 64.1 | 5.1 | 30.8 | |
| **Respondents can ask partner to use condom** | | | | | | <0.001 |
| No | 4,558 | 49.0 | 86.7 | 1.2 | 12.0 | |
| Yes | 4,183 | 44.9 | 68.0 | 2.5 | 29.5 | |
| Don't know | 565 | 6.1 | 87.1 | 1.4 | 11.5 | |
| **Ever been tested for HIV** | | | | | | |
| No | 9134 | 66.1 | 75.0 | 3.5 | 21.5 | <0.001 |
| Yes | 4679 | 33.9 | 55.7 | 2.4 | 41.9 | |
| **Wealth index** | | | | | | <0.001 |
| Poorest | 3,665 | 22.3 | 78.7 | 2.0 | 19.3 | |
| Poorer | 3,789 | 23.0 | 75.5 | 2.1 | 22.5 | |
| Middle | 3,560 | 21.6 | 71.4 | 3.3 | 25.3 | |
| Richer | 3,137 | 19.1 | 67.3 | 3.6 | 29.2 | |

*(Continued)*

**Table 2.** (Continued)

| Variables | Weighted (N = 16,442) | | Contraceptive Method Type | | | |
|---|---|---|---|---|---|---|
| | N | % | None (%) | Traditional (%) | Modern (%) | p-values |
| Richest | 2,291 | 13.9 | 60.6 | 4.6 | 34.8 | |
| **Type of residence** | | | | | | <0.001 |
| Urban | 5,042 | 30.7 | 63.1 | 4.4 | 32.5 | |
| Rural | 11,400 | 69.3 | 75.5 | 2.3 | 22.2 | |

Source: DHS 2012-2021

development of a National Sexual Health Strategy for Adolescents and Youth by the government of Chad in 2018 [29, 30]. According to the World Health Organization, 22.5% of married and 69.5% of unmarried adolescent females in Chad have unmet contraceptives needs.

In Namibia, however, government commitment in collaboration with international agencies played a critical role in encouraging contraceptive usage among adolescents [31]. For example, in 2020, the Namibian government in collaboration with the United Nations Population Fund (UNFPA) released revised National Guidelines for Family Planning. They pledged to invest in adolescents' youth-friendly reproductive health, and thus provided free family planning services at all public health facilities [31]. The fact that 23 out of the 25 SSA countries studied, or 92% of the sampled countries, had less than 50% contraceptive use among adolescent girls who were sexually active confirms the assertion that adolescents in SSA continue to face significant sexual and reproductive health challenges [26]. Similarly, a previous study of adolescent girls in 32 SSA countries that did not specifically target sexually active adolescent girls found an overall low prevalence of contraceptive use of 18.9% [32]. Another study conducted in 26 SSA found low (22.6%) prevalence of contraceptive use among adolescent girls in general, with no specificity for sexually active adolescent girls [33]. Low prevalence of contraceptive use among girls and women in SSA, particularly in Chad, is linked to socio-cultural and religious norms, preventing girls from accessing reproductive health services [30].

In this study, sexually active adolescent girls with primary or higher education are more likely to use contraceptives than those with no formal education. Previous studies in SSA countries such as Zambia, the Democratic Republic of the Congo, and Nigeria found a link between educational level and contraceptive use among adolescents [18, 20, 34]. Education boosts adolescents' self-esteem, confidence, reproductive health rights, and willingness to use contraceptives in order to avoid unwanted pregnancy in Benin and South Africa [22, 23]. The study established that married adolescent girls are less likely to use contraceptives than unmarried adolescent girls. Adolescent girls in SSA countries such as Nigeria, Zambia, and Ethiopia are confronted by cultural practices such as emphasis on virginity of girls, early child marriage, and encouragement of large family sizes. As a result, married adolescents in these countries are less likely to use contraceptives as they may prioritise getting pregnant in line with the prevailing cultural norms [10, 35–37]. Past studies noted that contraception is thought to encourage promiscuity in adolescent girls, thus making husbands to discourage their younger wives from using contraceptives [34, 38].

Findings also reveal that having many children is associated with less contraceptive use. Adolescent girls who have an ideal number of 6 or more children, for example, are less likely to use contraceptives than those who have an ideal number of 0 to 2 children, as they prioritise expanding their family. Previous studies in SSA found that cultural beliefs and a desire for a large family size discouraged married girls from using contraceptives [24, 39]. A similar study in SSA discovered that adolescent girls with ideal family sizes of six or more children were less

**Table 3. Binary logistic regression of contraceptive use among adolescent girls in sub-Saharan Africa.**

| Variables | Use no method | Use any method | Unadjusted Odds Ratio (95% CI) | Adjusted Odds Ratio (95% CI) |
|---|---|---|---|---|
| | n (%) | n (%) | | |
| **Education** | | | | |
| No education | | | Ref | Ref |
| Primary | | | 4.22 (3.76-4.73)*** | 2.65 (2.21-3.18)*** |
| Secondary | | | 7.57 (6.77-8.48)*** | 3.39 (2.68-4.03)*** |
| Higher | | | 8.79 (7.36-9.81)*** | 7.97 (6.26-9.45)*** |
| **Marital status** | | | | |
| Never in union | 3079 (56.3) | 2390 (43.7) | Ref | Ref |
| Ever married | 8705 (79.3) | 2268 (20.7) | 0.34 (0.32-0.37)*** | 0.34 (0.31-0.36)*** |
| **Ideal number of children** | | | | |
| 0-2 | 1137 (56.9) | 860 (43.1) | Ref | Ref |
| 3-5 | 5635 (64.3) | 3130 (35.7) | 0.80 (0.68-0.96)*** | 0.83 (0.69-0.98)* |
| 6+ | 4402 (87.8) | 611 (12.2) | 0.32 (0.26-0.39)*** | 0.30 (0.24-0.37)*** |
| **Total children ever born** | | | | |
| None | 7029 (75.5) | 2278 (24.5) | Ref | Ref |
| 1 | 3696 (64.3) | 2052 (35.7) | 8.58 (7.31-10.06)*** | 8.41 (7.19-9.83)*** |
| 2 or more | 1059 (76.4) | 328 (23.6) | 7.37 (5.97-9.11)*** | 6.91 (5.58-8.56)*** |
| **Ever had a terminated pregnancy** | | | | |
| No | 10900 (71.3) | 4395 (28.7) | Ref | Ref |
| Yes | 878 (77.0) | 263 (23.0) | 0.74 (0.64-0.86)*** | 0.89 (0.71-1.13) |
| **Heard family planning on radio last few months** | | | | |
| No | 9013 (74.0) | 3164 (26.0) | Ref | Ref |
| Yes | 2768 (65.0) | 1490 (35.0) | 1.37 (1.18-1.58)*** | 1.25 (1.09-1.44)** |
| **Heard family planning on TV last few months** | | | | |
| No | 10636 (73.2) | 3897 (26.8) | Ref | Ref |
| Yes | 1147 (60.2) | 757 (39.8) | 1.80 (1.63-1.99)*** | 0.80 (0.65-0.99) |
| **Heard family planning in newspaper/magazine last few months** | | | | |
| No | 11339 (72.6) | 4289 (27.4) | Ref | Ref |
| Yes | 445 (54.7) | 369 (45.3) | 2.19 (1.90-2.19)*** | 1.43 (1.04-1.96)* |
| **Respondents can ask partner to use condom** | | | | |
| No/Don't know | 4446 (86.8) | 678 (13.2) | Ref | Ref |
| Yes | 2844 (68.0) | 1338 (32.0) | 1.78 (1.56-2.02)*** | 1.71 (1.50-1.94)*** |
| **Ever been tested for HIV** | | | | |
| No | 6852 (75.0) | 2282 (25.0) | Ref | Ref |
| Yes | 2605 (55.7) | 2073 (44.3) | 2.21 (1.94-2.51)*** | 2.12 (1.86-2.41)*** |
| **Wealth index** | | | | |
| Poorest | 2886 (78.7) | 779 (21.3) | Ref | Ref |
| Poorer | 2860 (75.5) | 929 (24.5) | 1.18 (0.99-1.40)*** | 1.19 (1.00-1.42)* |
| Middle | 2540 (71.4) | 1019 (28.6) | 1.21 (1.01-1.45)* | 1.22 (1.02-1.47)* |
| Richer | 2110 (67.2) | 1028 (32.8) | 1.34 (1.09-1.64)** | 1.36 (1.11-1.66)** |
| Richest | 1388 (60.6) | 903 (39.4) | 1.89 (1.48-2.43)*** | 1.23 (1.73-2.88)*** |
| **Type of residence** | | | | |
| Urban | 3181 (63.1) | 1861 (36.9) | Ref | Ref |
| Rural | 8603 (75.5) | 2797 (24.5) | 1.40 (1.18-1.65)*** | 1.53 (1.29-1.81)*** |
| Model chi-square | | | 1959.819*** | 1915.452*** |
| -2 Log Likelihood | | | 6508.994 | 6503.075 |
| Cox & Snell $R^2$ | | | 0.231 | 0.231 |

*(Continued)*

**Table 3.** (Continued)

| Variables | Use no method | Use any method | Unadjusted Odds Ratio (95% CI) | Adjusted Odds Ratio (95% CI) |
|---|---|---|---|---|
| | n (%) | n (%) | | |
| Nagelkerke $R^2$ | | | 0.340 | 0.337 |
| Classification overall % (correct) | | | 79.7 | 78.8 |
| N | 16,442 | 16,442 | 16,442 | 16,442 |

Significant at p < .05

*, p < .01

**, p < .001

***; Ref – reference category; CI – confidence interval; n – weighted count; % - weighted percentage; DHS 2012-2021

likely to use contraceptives [24]. This study's findings also have confirmed that adolescent girls with two or more children are more likely to use contraceptives than those without a child. Other studies confirmed that the desire to avoid another childbearing experience or an unplanned pregnancy prompted girls to use contraceptives [40, 41]. Furthermore, counseling obtained during antenatal and postnatal examinations provided some adolescent mothers with knowledge and access to family planning in order to avoid unplanned pregnancy [42, 43].

It is found that adolescent girls who heard about family planning on the radio or read from the newspaper/magazine are more likely to use contraceptives than those who did not. The findings corroborate with previous findings from studies in Sierra Leone, Ghana, and South Asian countries regarding how exposure to family planning through the broadcast media, such as radio and newspaper, improves adolescents' contraceptive use [21, 44, 45]. Radio is an important media source for disseminating family planning information to adolescent girls in rural areas and to groups of young people who are not literate enough to read and understand. This has been made possible due to local radio stations that broadcast in indigenous languages [46, 47]. Newspapers and magazines are also important communication channels for educating the young population about family planning [21]. The texts and diagrams used to convey messages aid in the retention of family planning information, particularly about various methods of contraception, in the minds of the readers.

It is also discovered that adolescent girls who could request that their partners use a condom are more likely to use contraceptives than those who could not. The ability to request that a partner use a condom indicates that a girl has autonomy and control over her SRH issues. This finding substantiates that female decision-making power over SRH promotes sexual equality and helps to prevent unplanned pregnancy, as found by prior studies [32, 44]. It is also observed that girls who have been tested for HIV are more likely to use contraceptives than those who have never been tested. Possible explanation is that a girl who has been tested for HIV could have got the opportunity to receive counseling and information about the importance of abstinence or the use of condoms to prevent sexually transmitted infections, which is a means of encouraging the use of contraceptives [10].

It is also found that adolescent girls from other well-to-do and common households are more likely to use contraceptives than adolescent girls from the most impoverished households. Adolescent girls from wealthier households use contraceptives more frequently because they have enough money to cover the cost of contraception. This is further supported by previous findings that female adolescents in the richest wealth quintile were more likely to use contraception than female adolescents in the poorest wealth quintile in South Asian and SSA countries [21, 32, 45]. In comparing the use of modern and traditional methods, it is discovered that adolescent girls in rural areas are more likely than those in urban areas to use any

contraception methods. However, adolescent girls in urban areas are usually accustomed to the use of modern contraception than those in rural areas. This finding corresponds with several earlier studies in SSA indicating that adolescents in urban areas made use of modern contraceptives more than their counterparts in rural areas [10, 22, 24].

## Strengths and limitations

The study's strength is that it uses standardized and validated nationally representative datasets from 25 SSA countries, thus improving the generalizability of the study's findings to many African societies. Furthermore, the study used the most recent DHS datasets from the selected SSA countries, so the findings reflect current realities on the continent. Nonetheless, the study has some limitations. First, the utilized data were from cross-sectional surveys, indicating gathered information from respondents at a specific point in time, thus making longitudinal behavioral changes among adolescent girls difficult to obtain. In addition, because the data used were from cross-sectional surveys, the analysis could not indicate a causal-effect relationship, which is common in experimental research. The generalization of the study's findings across the entire SSA should be done with caution because the study only covered 25 countries in SSA and did not include the entire region. Furthermore, the adoption of the most recent DHS datasets from the selected countries may have caused differences in years of surveys, which could influence findings, particularly on prevalence of contraceptive use across countries in SSA, as recent surveys due to modernization and policy improvement in some countries may favour more contraceptive use among young people. Finally, because the surveys were conducted on human subjects and some interview questions may be retrospective in nature; there is possibility that the interpretation of findings may not adequately reflect current situation.

## Implications for research and policy

The study advocates for immediate policy action and efforts to mitigate the detrimental impacts of poor contraceptive use among sexually active adolescents in SSA. The prevalence of contraceptive use among sexually active adolescent females in SSA is as low as 25.4% on average, with variation across SSA nations. Chad, for example, has a prevalence of four percent. A closer look at our findings revealed that low contraceptive use among sexually active adolescent girls in SSA is linked to a lack of access to education, early marriage, a history of a terminated pregnancy, a lack of access to mass media, an inability to ask a partner to use a condom, and poverty. There is an urgent need for governments, civil society organizations, and nongovernmental organizations to take action to invest in girls' education in the SSA. All types of early marriage that expose young girls to early sexual debut must be prohibited by policy. This is crucial because adolescent girls who marry may lack the ability to make sexual and reproductive health decisions, such as asking their spouse to use a condom, due to their vulnerable age.

We advocate that government and relevant stakeholders invest in adolescents' reproductive health and make modern contraceptives available to them in order to prevent unplanned pregnancy and its associated consequences, such as school dropout, poverty, induced abortion, and pregnancy-related health complications. The media should be used to disseminate accurate information on reproductive health rights among adolescent females. Sexuality education should be promoted and included in school curricula. Adolescent girls, both in and out of school, should be given the opportunity to learn business skills in order to minimize abject poverty, which exposes disadvantaged adolescent girls to early sexual activity. Future research

should concentrate on the usage of contraception among sexually active adolescent boys in SSA.

## Conclusions

In general, a low prevalence of contraceptive use among sexually active adolescent girls in SSA has been established. As SSA countries strive to achieve the SDGs' target of universal access to SRH services, including family planning, and reducing unmet need for modern contraception, particularly among adolescent girls, it is critical for SSA countries with low contraceptive use to include unrestricted free access to family planning services as part of policy interventions to meet the contraceptives demands of its teeming young population. Both social media and mainstream media should be used to raise awareness about contraceptive use, using English, pidgin, and indigenous languages as communication tools to reach out to all types of young girls, including educated and uneducated, rich and poor, rural and urban dwellers. Efforts must also be made by policymakers to reduce child marriage in the SSA region, as adolescent girls who marry earlier miss out on formal education and are exposed to risky early motherhood due to a lack of empowerment for SRH rights.

## Supporting information

**S1 File. Country frequency statistics.**
(DOCX)

**S2 File. STROBE statement - checklist.**
(DOC)

**S3 File. VIF collinearity statistics.**
(DOCX)

**S4 File. Descriptive statistics SPSS output.**
(DOCX)

**S5 File. Logistic regression SPSS output.**
(DOCX)

## Acknowledgments

The authors are thankful to the DHS Program and all individuals involved in the study.

## Author Contributions

**Conceptualization:** Turnwait Otu Michael, Tolulope Funmilola Ojo, Olasupo Augustine Ijabadeniyi, Michael Ayodele Ibikunle, James Olukayode Oni, Adebanke Adeorite Agboola.

**Data curation:** Turnwait Otu Michael.

**Formal analysis:** Turnwait Otu Michael.

**Methodology:** Turnwait Otu Michael, Tolulope Funmilola Ojo.

**Supervision:** Turnwait Otu Michael, Tolulope Funmilola Ojo, Olasupo Augustine Ijabadeniyi, Michael Ayodele Ibikunle, James Olukayode Oni, Adebanke Adeorite Agboola.

**Visualization:** Turnwait Otu Michael, Tolulope Funmilola Ojo, Olasupo Augustine Ijabadeniyi, Michael Ayodele Ibikunle, James Olukayode Oni, Adebanke Adeorite Agboola.

**Writing – original draft:** Turnwait Otu Michael, Tolulope Funmilola Ojo, Olasupo Augustine Ijabadeniyi, Michael Ayodele Ibikunle, James Olukayode Oni.

**Writing – review & editing:** Turnwait Otu Michael, Tolulope Funmilola Ojo, Olasupo Augustine Ijabadeniyi, Michael Ayodele Ibikunle, James Olukayode Oni, Adebanke Adeorite Agboola.

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
