## [Decision Letter · Decision Letter 0]

18 May 2023

PONE-D-23-02018Prevalence and determinants of contraceptive use among sexually active adolescents girls in 25 sub-Saharan African countriesPLOS ONE

Dear Dr. Michael,

Thank you for submitting your manuscript to PLOS ONE. After careful consideration, we feel that it has merit but does not fully meet PLOS ONE’s publication criteria as it currently stands. Therefore, we invite you to submit a revised version of the manuscript that addresses the points raised during the review process.

We look forward to receiving your revised manuscript.

Kind regards,

Niharika Tripathi, PhD

Academic Editor

PLOS ONE

Journal Requirements:

Reviewers' comments:

Reviewer's Responses to Questions

**Comments to the Author**

1. Is the manuscript technically sound, and do the data support the conclusions?

Reviewer #1: Yes

Reviewer #2: Yes

Reviewer #3: Yes

2. Has the statistical analysis been performed appropriately and rigorously? 

Reviewer #1: Yes

Reviewer #2: Yes

Reviewer #3: No

3. Have the authors made all data underlying the findings in their manuscript fully available?

Reviewer #1: Yes

Reviewer #2: Yes

Reviewer #3: Yes

4. Is the manuscript presented in an intelligible fashion and written in standard English?

Reviewer #1: Yes

Reviewer #2: Yes

Reviewer #3: Yes

5. Review Comments to the Author

Reviewer #1: Prevalence and determinants of contraceptive use among sexually active adolescents girls in 25 sub-Saharan African countries

Previous studies

- A study with a similar objective was published see link - https://contraceptionmedicine.biomedcentral.com/articles/10.1186/s40834-020-00138-1

- Why there may be argument around this, it’s important that “modern contraceptive use” is also included in the overall “contraceptive use.”

Abstract

- Change all “determinants” to factors throughout the manuscript and title

- This statement, “However, due to a scarcity of multi-country empirical literature”, does not show the gap in knowledge whilst this study is important.

- Conclusion – not about this study examining access; why was “access” included in the conclusion? Free access might not be the issue, but other pertinent issues make recommendations in line with your study’s objective.

Introduction

- Without explicit information conviencing on why studies on modern contraceptive use among the same group in SSA is insufficient for better programmes and policies over contraceptive use, this study has not provided any knowledge gap. This argument could be the only way out of making this study unique. Why contraceptive use over modern contraceptive use despite the effective claim of modern contraceptive over other contraceptives as a whole, including “modern contraceptive”.

Method

- Well written

Results

- Figure 1 Remove “percentage”

- Write SSA in full in all the tables

- Provide table source

Discussion

- Provide more references to other studies conducted in Africa compared to your findings.

Other review

- Before the conclusion, include a section on the following;

1. Implication for Research and Policy

Reviewer #2: Thank you for the opportunity given to review that manuscript. This is an interesting study. However, I have a few comments which if the authors address will help strengthen the manuscript.

1. sub-Saharan Africa should be abbreviated in the abstract since that is the first use then “SSA” in the subsequent use

2. Authors should ensure that all the numbers that are not up 10 (eg: 9%, 4%) should be written in words/text

3. In the table 2, the authors used never in union in the presentation of the categories of marital status of the adolescents, that could give a confusing meaning/interpretation so, I suggest the author consider changing that or for example use “never married”

4. In the table 2, the authors using two categories in the presentation of the categories of educational level of the adolescents, for example the use of below secondary is ambiguous here, because there could be people among those group of adolescents who have never had any formal education and we cannot say per se that they had below secondary level education, so I suggest new categorisation is used that is more representative

5. In the table 2, the summation of the percentage should be check to ensure they all add up exactly to 100%, as it is now some are more than 100%

6. The report of independent variables in unadjusted odds ratios seem to be redundant with your previous report of all independent variable being significant in the report table 2

7. The authors report very low prevalence of contraceptive use in Chad and higher prevalence in Namibia, however, they fail to describe to the reader what Namibia is doing well (example: intervention) that might have contributed to the higher prevalence that need to be learned by other low prevalence countries

8. It will be helpful if the authors address grammatical errors and also the use of informal languages/English

Reviewer #3: Dear author(s),

Congratulations for putting this together. It was worth reading and reviewing this manuscript.

Please, I have made some useful comments that will improve the manuscript to be accepted for publication.

Kindly consider them for your inputs. Thank you

6. PLOS authors have the option to publish the peer review history of their article (what does this mean?). If published, this will include your full peer review and any attached files.

Reviewer #1: **Yes: **Obasanjo Afolabi Bolarinwa

Reviewer #2: No

Reviewer #3: No

---

## [Author Response · Author response to Decision Letter 0]

29 May 2023

Dear Editor, 

PLOS ONE Journal.

We thank you and the reviewers for your thoughtful comments on the manuscript, and we have updated it to address the issues raised. 

We have included a response table that lists all of the comments raised by reviewers and how we addressed them. 

We feel the work is now ready for your review and eventual publishing in PLOS ONE Journal. In addition also find below how we addressed the comments.

Thank you very much.

Turnwait O. Michael, PhD

On behalf of all authors.

Reviewer #1 Comments:

Comment: Prevalence and determinants of contraceptive use among sexually active adolescent girls in 25 sub-Saharan African countries

Previous studies

- A study with a similar objective was published see link - https://contraceptionmedicine. biomedcentral.com/articles/10.1186/s40834-020-00138-1

- Why there may be argument around this, it’s important that “modern contraceptive use” is also included in the overall “contraceptive use.”

Response: Thank you very much for identifying this study, and for bringing it to our notice. The article is referenced in our manuscript. The author, Ahinkorah (2020) focused on modern contraceptive use among adolescent girls (aged 15-19 years) and young women (aged 20-24 years). Our own study focuses primarily on sexually active adolescent girls (aged 15-19 years) who reported having been sexually active in the past four weeks before the survey. While Ahinkorah (2020) focused only on modern contraceptives, we focus on both modern and traditional contraceptive use. 

Comment: Change all “determinants” to factors throughout the manuscript and title

Response: Done; changed as suggested.

Comment: This statement, “However, due to a scarcity of multi-country empirical literature”, does not show the gap in knowledge whilst this study is important.

Response: Adjusted to now read “However, because contraceptive use among sexually active adolescents remains low and unexplored in sub-Saharan Africa, we examined…”

Comment: Without explicit information conviencing on why studies on modern contraceptive use among the same group in SSA is insufficient for better programmes and policies over contraceptive use, this study has not provided any knowledge gap. This argument could be the only way out of making this study unique. Why contraceptive use over modern contraceptive use despite the effective claim of modern contraceptive over other contraceptives as a whole, including “modern contraceptive”.

Response: It is no doubt that the modern contraceptive method is considered more effective than the traditional method of contraception. We attempted to understand the use of contraceptives by adolescent girls as a whole. In addition, what makes this study stand out is the focus which is on sexually active adolescents who had been sexually active in the previous four weeks before the survey. Meaning we were not just considering adolescents as a whole, but adolescent girls that had been sexually active in the last four weeks before the survey. 

Comment: Conclusion – not about this study examining access; why was “access” included in the conclusion? Free access might not be the issue, but other pertinent issues make recommendations in line with your study’s objective.

Response: Adjusted to now read “It is critical for the governments and civil societies in countries with low contraceptive use to strengthen mass education on the use of contraception among sexually active adolescents, with special emphasis on the less educated, married, and adolescent girls from poor households.”

Comment: Introduction

- Without explicit information conviencing on why studies on modern contraceptive use among the same group in SSA is insufficient for better programmes and policies over contraceptive use, this study has not provided any knowledge gap. This argument could be the only way out of making this study unique. Why contraceptive use over modern contraceptive use despite the effective claim of modern contraceptive over other contraceptives as a whole, including “modern contraceptive”.

Response: Thanks for this clarity. Sincerely, the debate on the effectiveness of modern contraceptive methods over other methods is inarguable. We captured this in the bivariate chi-square analysis. However, our study focused primarily on examining contraceptive use as a whole among sexually active adolescent girls. Not just among adolescent girls but sexually active adolescent girls who have been sexually active in the four weeks preceding the survey.

Comment: Method

- Well written

Response: Thank you

Comment: Results

- Figure 1 Remove “percentage”

Comment: Done; removed as suggested.

Comment: Write SSA in full in all the tables

Response: Done as suggested 

Comment: Provide table source.

Response: Done as suggested

Comment: Discussion

- Provide more references to other studies conducted in Africa compared to your findings.

Response: Done as suggested; now included references to studies conducted in Ghana, Nigeria, Sierra Leone, Burkina Faso, Ethiopia, South Africa, Namibia, Chad, etc.

Comment: Other review

- Before the conclusion, include a section on the following;

1. Implication for Research and Policy

Response: Done as suggested; now included “Implications for Research and Policy” section.

Reviewer #2 Comments

Comment: Thank you for the opportunity given to review that manuscript. This is an interesting study. However, I have a few comments which if the authors address will help strengthen the manuscript.

Response: Thank you.

Comment: 1. sub-Saharan Africa should be abbreviated in the abstract since that is the first use then “SSA” in the subsequent use

Response: Thank you greatly. We wish to do so as suggested, however, we could not because the PLOS ONE journal submission guideline for authors instructs that there should be no abbreviation in the abstract, if possible. 

Comment: 2. Authors should ensure that all the numbers that are not up 10 (eg: 9%, 4%) should be written in words/text

Response: Done as suggested 

Comment: 3. In the table 2, the authors used never in union in the presentation of the categories of marital status of the adolescents, that could give a confusing meaning/interpretation so, I suggest the author consider changing that or for example use “never married”

Comment: Done as suggested, now have “never married”

Comment: 4. In the table 2, the authors using two categories in the presentation of the categories of educational level of the adolescents, for example the use of below secondary is ambiguous here, because there could be people among those group of adolescents who have never had any formal education and we cannot say per se that they had below secondary level education, so I suggest new categorisation is used that is more representative

Response: Great suggestion. Thank you very much. We have now recategorized as suggested into: no education = 0, primary education = 1, secondary education = 2, and higher education = 3

Comment: 5. In the table 2, the summation of the percentage should be check to ensure they all add up exactly to 100%, as it is now some are more than 100%

Response: Done as suggested

Comment: 6. The report of independent variables in unadjusted odds ratios seem to be redundant with your previous report of all independent variable being significant in the report table 2

Response: Thank you very much for this observation. We didn’t check well to add the significant signs. It is now been corrected. Thank you!!

Comment: 7. The authors report very low prevalence of contraceptive use in Chad and higher prevalence in Namibia, however, they fail to describe to the reader what Namibia is doing well (example: intervention) that might have contributed to the higher prevalence that need to be learned by other low prevalence countries

Response: Done as suggested with examples of interventions/policy options. 

Comment: 8. It will be helpful if the authors address grammatical errors and also the use of informal languages/English

Response: Addressed and improved as suggested 

Reviewer #3 Comments:

Dear author(s),

Congratulations for putting this together. It was worth reading and reviewing this manuscript.

Please, I have made some useful comments that will improve the manuscript to be accepted for publication.

Kindly consider them for your inputs. Thank you

Response: Thank you greatly for your input. 

Comment: Conclusions -edit

Response: Corrected as suggested, now reads: “Conclusion.

Comment: Who in these countries should do this. It is always good if your recommendations are targeted towards a particular institution or institutions for action. 

Response: Done as suggested; it now reads: “It is critical for the governments and civil societies in countries with low contraceptive use to strengthen mass education on the use of contraception among sexually active adolescents, with special emphasis on the less educated, married, and adolescent girls from poor households.”

Comment: “While 43 births per 100 girls occurred worldwide in 2021, approximately 12 million of these births occurred in developing countries”

Dear authors, why are you mixing rates with figures? Please choose one and still to that particularly when doing comparison.

Response: Thanks a lot for this observation. It is corrected as suggested. It now reads: “As of 2021, around 12 million births occurred among adolescents in developing countries”

Comment: “Adolescents who stated that they had not been sexually active in the previous four weeks were excluded from the study” 

I suggest you delete this because it is repetition. Your immediate statement before this justifies your inclusion criterion.

Response: Done as suggested; now deleted. 

Comment: “Note: *DR = Democratic Republic; - = Not applicable”

What does this mean? You said you analysed data from 25 countries and by this table (Table 1), DR Congo is part. So, kindly explain what the Not applicable meant here. Thank you” 

Response: Thank you for your observation. The “- = Not applicable” refers to the dash (-) symbol in the body of Table 1, under the column with Year, at the point of All countries. We were not referring to DR Congo. However, to avoid further confusion for readers, we have now deleted the statement “- = Not applicable”. 

Comment: Did you combine no education and primary education as below secondary? If yes, this is not a good choice because clearly, we expect some variation in the use of contraceptives among those who have had at least primary education. How do we see the effect of education on 

Response: Thank you greatly. We have now recategorized as suggested into: no education = 0, primary education = 1, secondary education = 2, and higher education = 3

Comment: “To avoid confounding effects of explanatory variables, non-sexually active adolescent girls, as well as missing and non-responses, were dropped from the analysis”

This has already been said earlier. Please delete to avoid repetition

Response: Done as suggested; now deleted.

Comment: On education, reconsider the categorisation and run the analysis again. As it stands, it does not give indepth picture. As I said earlier, you cannot combine no education with primary education to = below secondary. This is misleading and entirely unacceptable.

Response: Thanks again for this observation. We have now recategorized as suggested into: no education, primary education, secondary education, and higher education. The analysis has also been run on this. 

Comment: This paragraph must be revised after recording the analysis. Referring to the paragraph on education under discussion section.

Response: Done as suggested; paragraph revised.

Journal Requirements:

Comment: When submitting your revision, we need you to address these additional requirements.

Response: Done, the manuscript meets PLOS ONE's style requirements, including those for file naming. 

Comment: 2. Please provide additional details regarding participant consent. In the ethics statement in the Methods and online submission information, please ensure that you have specified (1) whether consent was informed and (2) what type you obtained (for instance, written or verbal, and if verbal, how it was documented and witnessed). If your study included minors, state whether you obtained consent from parents or guardians. If the need for consent was waived by the ethics committee, please include this information.

Response: Done, the ethical section now reads:

“The authors obtained written authorization from the DHS Program to use DHS datasets in this study. The DHS received ethical approval from the ICF Institutional Review Board (ICF IRB FWA00000845) in the United States, as well as from the National Health Research Ethics Committees of the countries chosen for this study. In Nigeria, for example, the National Health Research Ethics Committee of Nigeria granted ethical approval for the survey (NHREC/01/01/2007). All participants provided written informed consent. In the case of minors under the age of 18, their parents or guardians provided informed consent. There was no need for further participation consent in this study because the research used DHS (secondary) datasets. More information about DHS ethical concerns can be found at https://goo.gl/ny8T6X. The DHS datasets are publicly accessible at https://dhsprogram.com/data/.”

Comment: 3. Please include captions for your Supporting Information files at the end of your manuscript, and update any in-text citations to match accordingly.

Response: Done accordingly

---

## [Decision Letter · Decision Letter 1]

9 Nov 2023

PONE-D-23-02018R1Prevalence and factors associated with contraceptive use among sexually active adolescent girls in 25 sub-Saharan African countriesPLOS ONE

Dear Dr. Michael,

Thank you for submitting your manuscript to PLOS ONE. After careful consideration, we feel that it has merit but does not fully meet PLOS ONE’s publication criteria as it currently stands. Therefore, we invite you to submit a revised version of the manuscript that addresses the points raised during the review process.

We look forward to receiving your revised manuscript.

Kind regards,

José Antonio Ortega, Ph.D.

Academic Editor

PLOS ONE

Journal Requirements:

**Additional Editor Comments:**

Note that the 2 reviewers recommend acceptance although it is suggested to copy edit the manuscript.

In addition, as the new academic editor I have inspected the manuscript. In particular, all the mentions to "Contraceptive method knowledge" should be removed from the table and from the text. It does not make much sense to include it as a regressor and it is not included in the final model probably due to lack of balance, since no women not knowing any method was a contraceptive user. No mention is made of the reason why it was dropped in the text. It should never have been included among the potential covariates since how could a women that declares not knowing any method report using a method?

Reviewers' comments:

Reviewer's Responses to Questions

**Comments to the Author**

1. If the authors have adequately addressed your comments raised in a previous round of review and you feel that this manuscript is now acceptable for publication, you may indicate that here to bypass the “Comments to the Author” section, enter your conflict of interest statement in the “Confidential to Editor” section, and submit your "Accept" recommendation.

Reviewer #1: (No Response)

Reviewer #3: All comments have been addressed

2. Is the manuscript technically sound, and do the data support the conclusions?

Reviewer #1: (No Response)

Reviewer #3: Yes

3. Has the statistical analysis been performed appropriately and rigorously? 

Reviewer #1: (No Response)

Reviewer #3: Yes

4. Have the authors made all data underlying the findings in their manuscript fully available?

Reviewer #1: (No Response)

Reviewer #3: Yes

5. Is the manuscript presented in an intelligible fashion and written in standard English?

Reviewer #1: (No Response)

Reviewer #3: Yes

6. Review Comments to the Author

Reviewer #1: All comments have been addressed. All comments have been addressed. All comments have been addressed.

Reviewer #3: Dear Authors,

Thank you for addressing all my comments adequately. However, I suggest that you do a proof reading one more time for some minor grammatical errors.

7. PLOS authors have the option to publish the peer review history of their article (what does this mean?). If published, this will include your full peer review and any attached files.

Reviewer #1: **Yes: **Obasanjo Bolarinwa

Reviewer #3: No

---

## [Author Response · Author response to Decision Letter 1]

30 Nov 2023

Comment: Note that the 2 reviewers recommend acceptance although it is suggested to copy edit the manuscript.

Response: Thank you greatly for this recommendation. The manuscript is now copy edited as suggested.

Comment: In addition, as the new academic editor I have inspected the manuscript. In particular, all the mentions to "Contraceptive method knowledge" should be removed from the table and from the text. It does not make much sense to include it as a regressor and it is not included in the final model probably due to lack of balance, since no women not knowing any method was a contraceptive user. No mention is made of the reason why it was dropped in the text. It should never have been included among the potential covariates since how could a woman that declares not knowing any method report using a method?

Response: Thank you very much for inspecting the manuscript and identifying this essential point. Contraceptive method knowledge is now removed from the table and from the text

---

## [Editor Report · Decision Letter 2]

14 Dec 2023

PONE-D-23-02018R2Prevalence and factors associated with contraceptives use among sexually active adolescent girls in 25 sub-Saharan African countriesPLOS ONE

Dear Dr. Michael,

Thank you for submitting your manuscript to PLOS ONE. After careful consideration, we feel that it has merit but does not fully meet PLOS ONE’s publication criteria as it currently stands. Therefore, we invite you to submit a revised version of the manuscript that addresses the points raised during the review process.

There was a title change from the previous version from "contraceptive use" to "contraceptives use". The international designation is "contraceptive use" so please go back to the earlier title and remove all instances (there are now many) of "contraceptives use".The description on p.6-7 on the outcome variable is unnecesarily lengthy. Since you are only considering the use of any method, just state so. You do not need to list and classify all methods. In fact you are not making any use of the question which method do you use. L. 214: "were twice as those who could not" missing "as likely". The same problem happens in the next lines, 215, 216, 217.L. 233: "The less prevalence 234 of contraceptives use " should be "the lower prevalence of contraceptive use" The interpretation of the lower prevalence of married adolescents, "Adolescent girls in SSA countries such as Nigeria, Zambia, and 265 Ethiopia are confronted by some cultural practices such as emphasizing virginity of girls, early 266 child marriage, and encouraging large family size. As a result, married adolescents are less 267 likely to use contraceptives in these countries" is missing an easier and more direct mechanism: married adolescents are less likely to use contraception because they (more of them) want to get pregnant. That is what contraceptives are for! "Findings also establish that having many children can be associated with less 271 contraceptives use": Note this can't be a causal effect. In fact, you use contracetion not to have children! It is rather a tell-tale sign that the particular woman does not want to use contraception or control her fertility. You should tone down this claim: "Because of their fragile age, adolescent girls who marry will undoubtedly lack the 20 347 ability to make sexual and reproductive health decisions such as asking their spouse to use a 348 condom. " You could just replace "will undoubtedly" with "might".In addition, please remove some recommendations that do not stem from the analysis such as:"Sexual intercourse between married adolescents and their partners is expected in most 350 societies that practise early marriage since it is regarded to be inside the marital bond [36, 37]. 351 This practice never took into account young girls' bodies and psychology, which are still 352 immature to sexual intercourse and the pregnancy repercussions"Note that no international criteria advocate universal lack of sexual activity in adolescents. You also should not say ". We advocate for sexual abstinence 354 for all adolescents".Remember what the international consensus is that adolescents have the same SRH rights as adults, that is, "Reproductive health implies that people are able to have a satisfying and safe sex life and that they have the capability to reproduce and the freedom to decide if, when and how often to do so.".

We look forward to receiving your revised manuscript.

Kind regards,

José Antonio Ortega, Ph.D.

Academic Editor

PLOS ONE
---

## [Author Response · Author response to Decision Letter 2]

19 Dec 2023

Comments: There was a title change from the previous version from "contraceptive use" to "contraceptives use". The international designation is "contraceptive use" so please go back to the earlier title and remove all instances (there are now many) of "contraceptives use".

Response: Thank you for bringing this important issue to our attention and providing clarification regarding the international designation of the term "contraceptive use." We have made the necessary correction, changing 'contraceptives use' to 'contraceptive use' in the title and throughout the manuscript.

Comment: The description on p.6-7 on the outcome variable is unnecessarily lengthy. Since you are only considering the use of any method, just state so. You do not need to list and classify all methods. In fact you are not making any use of the question which method do you use.

Response: We have removed the unnecessarily lengthy component of the outcome variable. It now reads “The use of contraceptives by adolescent girls who were sexually active was the study’s outcome variable. Female and male sterilization, IUD, injectables, implants, pill, condom, female and male condoms, diaphragm, foam/jelly, lactational amenorrhea, rhythm, withdrawal, no methods were the options available to respondents. To create a binary outcome variable for logistic regression, the contraceptive method categories were recoded as follows: no method = 0, any method= 1. 

Comment: L. 214: "were twice as those who could not" missing "as likely". The same problem happens in the next lines, 215, 216, 217.

Response: We have included the “as likely” in all the affected areas. The sections now read: Adolescent girls who could request that their partners use a condom were twice as likely as those who could not. Girls who had ever been tested for HIV were twice as likely as those who had never been tested to use contraceptives. Others were more likely to use contraceptives than adolescent girls from the poorest households. Surprisingly, adolescent girls in rural areas were twice as likely as those in urban areas to use contraceptives.

Comment: L. 233: "The less prevalence 234 of contraceptives use " should be "the lower prevalence of contraceptive use"

Response: Corrected. It now reads: The lower prevalence of contraceptive use…

Comment: The interpretation of the lower prevalence of married adolescents, "Adolescent girls in SSA countries such as Nigeria, Zambia, and 265 Ethiopia are confronted by some cultural practices such as emphasizing virginity of girls, early 266 child marriage, and encouraging large family size. As a result, married adolescents are less 267 likely to use contraceptives in these countries" is missing an easier and more direct mechanism: married adolescents are less likely to use contraception because they (more of them) want to get pregnant. That is what contraceptives are for!

Response: Corrected. This now reads: Adolescent girls in SSA countries such as Nigeria, Zambia, and Ethiopia are confronted by cultural practices such as emphasis on virginity of girls, early child marriage, and encouragement of large family sizes. As a result, married adolescents in these countries are less likely to use contraceptives as they may prioritise getting pregnant in line with the prevailing cultural norms [10, 35–37]

Comment: "Findings also establish that having many children can be associated with less 271 contraceptives use": Note this can't be a causal effect. In fact, you use contraception not to have children! It is rather a tell-tale sign that the particular woman does not want to use contraception or control her fertility.

Response: Corrected. This now reads: Findings also reveal that having many children is associated with less contraceptive use. Adolescent girls who have an ideal number of 6 or more children, for example, are less likely to use contraceptives than those who have an ideal number of 0 to 2 children, as they prioritise expanding their family.

Comment: You should tone down this claim: "Because of their fragile age, adolescent girls who marry will undoubtedly lack the 20 347 ability to make sexual and reproductive health decisions such as asking their spouse to use a 348 condom. " You could just replace "will undoubtedly" with "might".

Response: Corrected. This section now reads: All types of early marriage that expose young girls to early sexual debut must be prohibited by policy. This is crucial because adolescent girls who marry may lack the ability to make sexual and reproductive health decisions, such as asking their spouse to use a condom, due to their vulnerable age. 

Comment: In addition, please remove some recommendations that do not stem from the analysis such as:

"Sexual intercourse between married adolescents and their partners is expected in most 350 societies that practise early marriage since it is regarded to be inside the marital bond [36, 37]. 351 This practice never took into account young girls' bodies and psychology, which are still 352 immature to sexual intercourse and the pregnancy repercussions"

Response: The highlighted section is removed from the manuscript. 

Comment: Note that no international criteria advocate universal lack of sexual activity in adolescents. You also should not say ". We advocate for sexual abstinence 354 for all adolescents".

Remember what the international consensus is that adolescents have the same SRH rights as adults, that is, "Reproductive health implies that people are able to have a satisfying and safe sex life and that they have the capability to reproduce and the freedom to decide if, when and how often to do so.".

Response: Thanks for this insight. We have adjusted our advocacy statement to now read: We advocate that government and relevant stakeholders invest in adolescents' reproductive health and make modern contraceptives available to them in order to prevent unplanned pregnancy and its associated consequences, such as school dropout, poverty, induced abortion, and pregnancy-related health complications.

---

## [Editor Report · Decision Letter 3]

4 Jan 2024

Prevalence and factors associated with contraceptive use among sexually active adolescent girls in 25 sub-Saharan African countries

PONE-D-23-02018R3

Dear Dr. Michael,

We’re pleased to inform you that your manuscript has been judged scientifically suitable for publication and will be formally accepted for publication once it meets all outstanding technical requirements.

Kind regards,

José Antonio Ortega, Ph.D.

Academic Editor

PLOS ONE

Additional Editor Comments (optional):

The changes respond to the issues raised by the editor, the reviewers were already advising to accept in earlier versions. The work is ready for publication.
---

## [Editor Report · Acceptance letter]

17 Feb 2024

PONE-D-23-02018R3 

PLOS ONE

Dear Dr. Michael, 

I'm pleased to inform you that your manuscript has been deemed suitable for publication in PLOS ONE. Congratulations! Your manuscript is now being handed over to our production team.

Kind regards, 

on behalf of

Dr. José Antonio Ortega 

Academic Editor

PLOS ONE